# Phosphoproteome Profiling of uEVs Reveals p-AQP2 and p-GSK3β as Potential Markers for Diabetic Nephropathy

**DOI:** 10.3390/molecules28145605

**Published:** 2023-07-24

**Authors:** Qing Li, Jiong Zhang, Yi Fang, Yan Dai, Ping Jia, Ziyan Shen, Sujuan Xu, Xiaoqiang Ding, Feng Zhou

**Affiliations:** 1Department of Nephrology, Zhongshan Hospital, Fudan University, Shanghai 200437, China; faith0113@126.com (Q.L.); fang.yi@zs-hospital.sh.cn (Y.F.); daishu1996@126.com (Y.D.); jia.ping1@zs-hospital.sh.cn (P.J.); shen.ziyan@zs-hospital.sh.cn (Z.S.); 2Department of Nephrology, Sichuan Academy of Sciences & Sichuan Provincial People’s Hospital, Sichuan Clinical Research Center for Kidney Disease, University of Electronic Science and Technology, Chengdu 610072, China; zhangjiong831224@163.com; 3Department of Nephrology, The Third Hospital of Hebei Medical University, Shijiazhuang 050051, China; 15810500570@163.com; 4Key Laboratory of Carcinogenesis and Cancer Invasion, Liver Cancer Institute, Zhongshan Hospital, Minister of Education, Institutes of Biomedical Sciences, Fudan University, Shanghai 200032, China

**Keywords:** phosphoproteome, urinary extracellular vesicles, diabetic nephropathy, biomarker

## Abstract

Diabetic nephropathy (DN) contributes to increased morbidity and mortality among patients with diabetes and presents a considerable global health challenge. However, reliable biomarkers of DN have not yet been established. Phosphorylated proteins are crucial for disease progression. However, their diagnostic potential remains unexplored. In this study, we used ultra-high-sensitivity quantitative phosphoproteomics to identify phosphoproteins in urinary extracellular vesicles (uEVs) as potential biomarkers of DN. We detected 233 phosphopeptides within the uEVs, with 47 phosphoproteins exhibiting significant alterations in patients with DN compared to those in patients with diabetes. From these phosphoproteins, we selected phosphorylated aquaporin-2 (p-AQP2[S256]) and phosphorylated glycogen synthase kinase-3β (p-GSK3β[Y216]) for validation, as they were significantly overrepresented in pathway analyses and previously implicated in DN pathogenesis. Both phosphoproteins were successfully confirmed through Phos-tag western blotting in uEVs and immunohistochemistry staining in kidney sections, suggesting that phosphoprotein alterations in uEVs reflect corresponding changes within the kidney and their potential as candidate biomarkers for DN. Our research proposes the utilization of phosphoproteins in uEVs as a liquid biopsy, presenting a highly feasible diagnostic tool for kidney disease.

## 1. Introduction

Diabetic kidney disease (DKD) has emerged as one of the fastest-growing causes of chronic kidney disease [1]. The clinical diagnosis of DKD includes three possibilities: diabetic nephropathy (DN), non-diabetic kidney disease (NDKD), and a combination of DN and NDKD. The treatment and prognosis of DN and NDKD differ significantly [2]. DN diagnosis relies on renal biopsy, an invasive procedure with potential risks, such as bleeding and kidney damage, which is not widely practiced and is unsuitable for monitoring treatment responses. Consequently, most cases of DN are clinically diagnosed. The primary laboratory markers for the clinical diagnosis of DN include albuminuria and reduced glomerular filtration rate. However, not all diabetes patients with abnormal laboratory markers have DN, and DN progression has been observed in the absence of these markers [3]. Given the complexities of DKD and the pressing need for more accurate and non-invasive early diagnostic biomarkers of DN, this area has experienced a surge in active research efforts [4,5].

The identification of biomarkers in extracellular vesicles (EVs) has sparked a rapidly growing scientific field and renewed interest in biomarker discovery. EVs contain proteins, miRNAs, and mRNAs that reflect the pathophysiological state of their originating cells and are a rich source of potential biomarkers [6]. Urinary extracellular vesicles (uEVs), which are actively secreted by nephron-lining epithelial cells, are considered more specific and potentially sensitive [7]. In particular, proteins provide more direct information about disease progression than RNA. Proteomics may serve as a valuable tool for identifying novel uEVs biomarkers. Large-scale proteomic assessments supported the reliability of monitoring uEVs protein changes as a proxy for kidney alterations [8]. Additionally, uEVs represent kidney protein modifications more accurately compared with the whole urine samples, which contain only a few kidney-derived proteins. Approximately 3% of the total urinary protein from healthy subjects originates from uEVs, yielding a more than 30-fold enrichment of proteins that are minor components of whole urine and are thus readily detectable by protein mass spectrometry [9]. In addition, uEVs substantially reduce sample complexity compared to whole urine, particularly in kidney disease, where a larger amount of protein may be present. Several studies have provided strong evidence that uEVs-derived disease markers can be identified well before symptom onset, making them promising candidates for early-stage disease detection [10]. Renal biopsy, the current diagnostic method for DN, is invasive, limited by the number of obtainable glomeruli and focal sites, and may be insufficient to evaluate the whole-kidney injury. In contrast, uEVs represent the collective response of different renal cells in all renal units to various pathological stimuli, offering the potential to obtain comprehensive kidney information noninvasively and facilitate follow-up. uEVs proteins are an ideal source of biomarkers for kidney diseases. Proteins such as podocalyxin, C-megalin, AQP2, and dipeptidyl peptidase-IV in uEVs have recently emerged as potential DN biomarker candidates in various studies [11,12,13]. Despite their high sensitivity, these proteins lack specificity for DN diagnosis, which is a major barrier to the development of new kidney disease biomarkers.

Phosphorylation events are considered critical factors in the early onset and progression of kidney disease [14,15], providing more direct insights into disease status than total protein levels. However, urine-derived phosphoproteins have been largely unexplored because of the presence of phosphatases in urine and the low abundance of phosphoproteins in highly complex urine samples. Intriguingly, uEVs are highly stable membrane-enclosed vesicles that protect their contents from external proteases and phosphatases [16], potentially enabling the development of uEVs phosphoproteins as diagnostic tools for kidney diseases. In recent years, disease-associated phosphorylated proteins in EVs have been demonstrated as disease markers [17,18]. However, these studies are predominantly hypothesis-based, which poses significant limitations. Few studies employing cellular or plasma EV phosphorylation proteomics for disease marker screening have confirmed the potential value of phosphorylated proteins in EVs as diagnostic disease markers [16,19,20]. Although phosphoproteins are present in low abundance, unbiased quantitative phosphoproteomic analysis for identifying uEVs phosphoproteins offers an innovative approach for renal disease diagnosis. In this study, we aimed to develop uEVs phosphoproteins as potential kidney disease biomarkers with a focus on DN.

## 2. Results

### 2.1. Clinical Characteristics of the Study Group

In this study, 15 male participants (*n* = 5 per group) were included, with no significant differences in sex or age, which are factors known to influence the type and number of uEVs [6]. Table 1 summarizes the clinical characteristics of the study groups. The stature and weight of the study groups were shown in Appendix A. As anticipated, both the albumin-to-creatinine ratio (ACR) and estimated glomerular filtration rate (eGFR) were significantly higher in the diabetic nephropathy (DN) group than in the diabetes (DM) group. Written informed consent was obtained from all participants, and the protocols were approved by the institutional ethics committee.

### 2.2. uEVs Isolation and Characterization

Gel electrophoresis of uEVs from NC subjects and DN patients showed that the coprecipitation of albumin with uEVs by ultracentrifugation could be removed by sucrose gradient density centrifugation (Figure 1a). Transmission electron microscopy (TEM) illustrated the morphology of the uEVs (Figure 1b). Nanoparticle Tracking Analysis (NTA) revealed a varied size distribution of uEVs, with a peak diameter of less than 100 nm (Figure 1c). Western blot analysis confirmed uEVs enrichment in the final fraction, based on the expression of extracellular vesicle markers Alix, TSG101, and CD63 (Figure 1d). The NTA curve represents the concentrations of components of various particle sizes in the sample. According to the particle size distribution and concentration changes in NTA, the particle size distribution of uEVs purified by sucrose gradient density centrifugation was simpler, and the particle size was smaller than that of uEVs purified by ultracentrifugation. This observation aligns with the TEM findings. We used Alix, TSG101, and CD63 as uEVs markers based on current research on uEVs [6]. All these markers were detected in uEVs isolated by both methods, with high expression (Figure 1c). Only TSG101 was detected in the urine supernatant, indicating a degree of uEVs loss during centrifugation. This also suggests that there is no specific marker for uEVs, necessitating the use of various methods to characterize them. Our results demonstrate the successful isolation of uEVs from urine through sucrose gradient density centrifugation.

### 2.3. Ultra-High Sensitivity Quantitative Phosphoproteomic Analysis

In this study, a phosphoprotein enrichment technique combined with iTRAQ labeling and quantitative tandem mass spectrometry was employed to investigate the phosphorylation levels of proteins in uEVs. A total of 2652 proteins and 233 unique phosphopeptides were identified in uEVs. The Progenesis QI software (version 2.3) was used for iTRAQ labeling quantification to determine the differential phosphorylation of uEVs proteins between patients with DM and DN.

This methodology identified 292 phosphorylation sites corresponding to 233 phosphorylated peptides and 214 phosphorylated proteins. The identified phosphorylation sites included 179 (61.3%) serine, 101 (34.6%) threonine, and 12 (4.11%) tyrosine phosphorylation sites (Figure 2). Among the phosphorylated proteins, 31 were expressed in all the groups, accounting for 14.49% of the total phosphorylated proteins. Previous large-scale phosphoproteomic studies suggested that phosphorylation predominantly occurs in nuclear proteins [21]. However, a significant portion of the uEVs phosphoproteome appears to originate from membranes and organelles. The results encompassed various high-abundance phosphorylated proteins, primarily associated with proteins involved in uEVs formation, structural proteins that mediate cell-cell interactions and signaling, glucose metabolism-related proteins, and an assortment of channel proteins, including ionic and aquaporin channels.

Phosphopeptides with a probability score of phosphorylation site location above 1.5 and *p* < 0.05, were identified as differential phosphorylation events in patients with DN compared to diabetic patients. Table 2 shows significant differences in uEVs protein phosphorylation between patients with DN and diabetes. The results revealed 23 upregulated and 24 downregulated phosphopeptides in patients with DN compared to those in patients with diabetes. However, differential phosphorylation may result from changes in protein expression and phosphorylation. Therefore, the total proteomes of uEVs were also analyzed.

A total of 2652 proteins were identified, of which 74% were also identified within the phosphoproteome and 26% were detected solely in the phosphorylation data, indicating that phosphoproteins are of considerably low abundance and may escape detection using the shotgun proteomics approach. The quantitative proteome analysis of uEVs demonstrated similar expression of most proteins in patients with DN and DM, whereas phosphopeptide levels were significantly different between patients with DN and those with diabetes. These findings suggest that the phosphorylation differences between patients with DN and DM are not a result of changes in protein expression and that phosphorylated proteins are more specific to DN patients (Figure 3). The results also validated our strategy to indicate disease progression through changes in phosphorylated proteins rather than changes in total protein expression.

### 2.4. Gene Ontology Analysis

Gene ontology analysis of the phosphoproteins indicated that the phosphorylated protein upregulated in the DN group belonged to “microtubule” and “microtubule-based movement” in the cell component clustering, “microtubule-based movement” in the biological process clustering, “microtubule binding” and “tubulin binding” in the molecular function clustering (Figure 4). Microtubules play a key role in regulating the secretion of insulin particles through their transport to the cell membrane. Glucose regulates microtubule function and promotes microtubule formation in the Golgi apparatus, thereby promoting insulin granule formation. The upregulation of microtubule clustering in DN may be due to prolonged hyperglycemia, resulting in compensatory upregulation of microtubule clustering and increased insulin secretion [22].

### 2.5. Protein-Protein Interaction Prediction of the Phosphoproteins in uEVs from Diabetic Patients Compared to Patients with DN

To gain a further understanding of the functional interactions among the significantly altered phosphoproteins between patients with DN and DM, a protein-protein interaction network was analyzed using STRING, setting the minimum interaction coefficient to an intermediate confidence level of 0.7. The connections are depicted in Figure 5, with different colors representing various biological processes in the clustering. The network demonstrated that several phosphoproteins are involved in cellular component organization and cellular processes. AQP2 interacts with the calcium channel protein TRPV5 and the sodium phosphate channel protein SLC34A1, both of which belong to the same channel protein interaction network. GSK3β and BAIAP2, which are associated with glycometabolism, shared the same interaction networks. GSK3β and AQP2 are key proteins in the interaction network.

### 2.6. Validation of Identified Phosphorylation Specific to DN Patients by Phos-Tag Western Blotting and Immunohistochemistry

The heterogeneity of DN renders it unlikely that a single diagnostic biomarker can be identified. However, multiple candidate markers that reflect the onset and progression of DN may offer improved diagnostic and prognostic values. To initially verify candidate phosphoproteins in patients with DN, immunodetection was employed to validate the distinct phosphorylated proteins, including p-AQP2 and p-GSK3β. Given the considerable variation in uEVs numbers among clinical samples, which can significantly affect the expression of phosphorylated proteins, the normalization of uEVs in all samples is necessary. Alix, a marker protein of uEVs, was used for normalization [22,23]. By normalizing uEVs with phosphorylated protein/Alix, phosphorylated protein expression in the same number of uEVs could be compared, avoiding differential phosphorylation events due to differences in the number of uEVs. Western blot analyses revealed that p-AQP2 in uEVs was significantly downregulated in patients with DN compared to diabetic patients, while p-GSK3β in uEVs was significantly upregulated in patients with DN compared to diabetic patients (Figure 6). These results are consistent with previous findings and indicate that phosphorylated proteins are more prevalent in uEVs than in urine.

A large-scale proteomic assessment of urinary extracellular vesicles has recently demonstrated their reliability in reflecting changes in kidney protein levels [8,24]. To confirm the hypothesis that phosphoproteins in uEVs can also represent phosphorylation events in kidney tissues, IHC was performed on kidney tissues from mouse models and patients with DN. The well-established animal models for type 2 diabetes and DN, db/db mice, exhibit insulin resistance and hyperglycemia in the first 8 w of life and are characterized by progressive albuminuria and significant matrix accumulation in the glomeruli from 28 w of age. By contrast, db/db mice at 8 w of age and control wild-type mice (db/m) exhibited no renal disease. Based on the phosphoprotein analysis and immunodetection results, pAQP2(S256) and GSK3β(Y216) were investigated in db/m and db/db mice at 8 and 28 w of age. The renal expression of p-GSK3β(Y216) in db/db mice at 28 w of age was higher than that in db/db mice at 8 w of age, whereas that of p-AQP2(S256) decreased. Both showed similar changes in the kidney tissue of DN patients compared to those of DM patients with NDKD (Figure 7).

## 3. Discussion

Phosphorylated proteins are involved in human diseases, play critical roles in disease progression, and specifically indicate pathological states. Although phosphorylated proteins have been known for over a century, their low abundance, wide dynamic range, and complexity and diversity of clinical samples have consistently posed significant challenges to their large-scale assessment. Despite advancements in mass spectrometry to improve the discovery of proteins, data quality is inversely proportional to sample complexity.

Urine is the primary sample source for diagnosing kidney diseases; however, it contains a few phosphorylated proteins owing to the presence of phosphatases. A recent urinary phosphoproteomic study identified only 64 phosphoproteins [25,26]. Moreover, the urine of patients with kidney disease often contains large amounts of proteins from the circulating plasma, particularly in patients with DN, making the detection of small amounts of phosphoproteins even more challenging. We separated and purified the uEVs using sucrose gradient density centrifugation to minimize the effects of large amounts of urine protein in patients with DN. The isolation of uEVs from urine and the enrichment of phosphorylated proteins substantially reduces the complexity of clinical urine samples.

All nephron-lining epithelial cells actively secrete uEVs that are sensitive and specific to pathophysiological changes in the kidneys. The lipid membrane of uEVs provides stability and protects the phosphorylated proteins within from degradation by phosphatases, thereby obtaining more complete and reliable kidney information [27,28]. Previous studies have demonstrated that uEVs are an ideal source of kidney disease markers [29,30,31,32,33]. In this study, we identified phosphoproteins in uEVs as DN markers using quantitative phosphorylated proteomics with ultrahigh sensitivity. The use of high-throughput, ultra-high-sensitivity proteomic methods has enabled the mining of uEVs as biomarkers. The low abundance of phosphoproteins and phosphoryl bonds, which are easier to break than peptide bonds, makes the identification of phosphoproteins more challenging than that of proteins. Traditional proteomics struggles to effectively cover low-abundance phosphorylated proteins, necessitating high-sensitivity mass spectrometry. We further enhanced sensitivity by constructing a nanoscale full-online three-dimensional ultra-high-sensitivity proteomics platform and using isobaric tags for quantitation (iTRAQ) for biomarker discovery.

In electrospray mass spectrometry, a lower volumetric flow rate results in higher mass detection sensitivity. With a nanoscale volumetric flow rate upgrade, such as 1–3 nL/min, the detection sensitivity increased by an order of magnitude compared to 100–300 nL/min. However, at such low volumetric flow rates, the mobile phase moves below the velocity recommended by the Van Deemter curve, leading to a significant reduction in column resolution and affecting peptide separation [34,35]. The most effective solution to this problem is to use capillary columns with the smallest possible inner diameters. However, capillary columns with very small inner diameters tend to exhibit poor reliability, short lifetimes, and decreased maximum sample volumes; therefore, they are not widely used. We established long-column technology with a 25-µm inner diameter and integrated the precolumn technology to comprehensively address these issues. The sample was first adsorbed by the pre-column and then separated and analyzed in a long analytical column with an inner diameter of 25 µm, providing sufficient column capacity so that highly expressed peptides do not overload the analytical column. Simultaneously, a column length of 25-µm inner diameter and 100 cm maximizes the resolution of the separation column with a volumetric flow rate of 1–3 nL/min, thereby maximizing the ionization efficiency of the electrospray, achieving better peptide separation, and realizing the ultra-high-sensitivity detection of mass spectrometry.

Second, we employed a three-dimensional orthogonal basic reversed-strong anion exchange-acid reversed-phase chromatographic system (RP-SAX-RP). In chromatographic detection, high-expression peptides considerably affect low-expression peptides, such as phosphorylated peptides. The orthogonal multidimensional chromatographic separation of complex peptide mixtures effectively mitigates the interference from high-expression peptides, thus enhancing the detection of low-expression phosphorylated peptides. The three-dimensional chromatography system amalgamates fully orthogonal basic reversed-phase, strong anion exchange, and acidic reversed-phase chromatographic systems, achieving a peak capacity of approximately 12,000 for 20 components and establishing a robust foundation for detecting low copy number proteins [36,37].

Moreover, we constructed a fully online automatic analysis platform that significantly reduced sample loss during the transfer process by incorporating the aforementioned chromatographic platforms into a nano-upgrade-integrated all-online system. A fully automated system requires no intervention, maximizing the sensitivity of the entire system and enabling proteomics to reach a depth of determination similar to that of next-generation genome sequencing-based technology. Consequently, the coverage of phosphorylated proteins, which is crucial for functional regulation, has substantially improved, facilitating the quantitative detection of phosphorylated proteins in uEVs. Isotope labeling is currently the most accurate method for quantifying phosphorylated proteins. We substantially enhanced the accuracy of large-scale phosphorylated peptide quantification by introducing a two-isotope labeling internal reference. Previous non-quantitative phosphoproteomic studies of uEVs detected only 19 phosphorylation sites in healthy individuals, corresponding to 14 phosphorylated proteins [38]. This approach allowed us to identify 292 phosphorylation sites that matched 233 phosphorylated peptides and 214 phosphorylated proteins. These results indicated that our uEVs phosphoproteome was sufficiently sensitive to isolate and identify hundreds of phosphopeptides in uEVs.

We employed iTRAQ quantitative analysis to identify the phosphorylated proteins in the different groups. The analysis of KEGG enrichment results suggested that signaling pathways, such as water reabsorption regulated by arginine vasopressin (AVP), in patients with DN were significantly downregulated compared to those in patients with diabetes, while multiple glucose metabolism-related signaling pathways were significantly upregulated. Both p-GSK3β(Y216) and p-AQP2(S256) in uEVs were significantly higher in patients with diabetes than that in healthy controls. uEVs p-GSK3β(Y216) were significantly increased in patients with DN compared to diabetic patients, while uEVs p-AQP2(S256) were significantly decreased.

Diabetic nephropathy is associated with a severely disrupted water balance due to the abnormal regulation of renal aquaporins. AQP2, the most important aquaporin in urine concentration, is regulated by AVP and is located in the apical plasma membrane and vesicles of the collecting duct principal cells [39]. Studies have shown that p-AQP2(S256) plays a critical role in the AVP-induced transfer of AQP2 to the apical plasma membrane of the lumen in the collecting duct, promoting the reabsorption of water [40,41,42]. Our study found that the levels of uEVs p-AQP2(S256) were significantly higher in patients with diabetes than in healthy controls, which may be a compensatory mechanism to reabsorb water and mitigate the increased plasma osmolality caused by hyperglycemia. In contrast, uEVs p-AQP2(S256) was significantly decreased in patients with DN, which may be related to decreased renal function and impaired urine concentration in patients with DN.

Persistent hyperglycemia can lead to insulin resistance in type 2 diabetes, and previous studies have demonstrated that GSK-3β is strongly associated with the development of insulin resistance [43]. Our study found that uEVs p-GSK3β(Y216) were significantly increased in diabetic patients compared with healthy controls, consistent with previous findings. GSK-3β is also closely related to the cell signaling pathway involved in renal injury repair and regeneration and mediates podocyte damage [44]. Our study found that uEVs p-GSK3β(Y216) were significantly increased in patients with DN compared with diabetic patients. Moreover, in the kidney tissue of a mouse model of DN, both p-AQP2(S256) and p-GSK3β(Y216) were similarly altered, indicating their important roles in the progression of DN and their potential as biomarkers for the diagnosis of DN.

Strict inclusion and exclusion criteria were set for subjects in the discovery phase of biomarkers to ensure homogeneity within the group and contribute to the screening of ideal biomarkers. Our study included patients with a confirmed history of diabetes for less than two years and a normal blood glucose one year before the diagnosis of DM to exclude patients with pathological changes of DN but no clinical manifestations. The enrolled patients with DN had pathological grades III–IV. Although strict enrollment criteria ensure intragroup homogeneity and facilitate the discovery of ideal biomarkers, these markers must be further validated in heterogeneous populations.

The utilization of spot urine samples yields reproducible data and suggests the potential use of uEVs to diagnose renal disease [6]. However, sample preparation for uEVs poses additional challenges to the accuracy of MS-based targeted quantitation. Our study provides a new strategy for the isolation and identification of hundreds of phosphopeptides from uEVs, indicating their potential as liquid biopsies for the treatment of kidney disease. p-AQP2(S256) and p-GSK3β(Y216) are potential biomarkers for the diagnosis of DN, but further validation in a larger heterogeneous cohort of patients with DN is required.

## 4. Materials and Methods

### 4.1. Patient Recruitment

Patients with DN and Type 2 diabetes were recruited from the Zhongshan Hospital of Endocrinology and Nephrology affiliated with Fudan University. Healthy controls who had undergone a medical examination within the past year were selected. The inclusion criteria were patients with a history of type 2 diabetes of less than two years and kidney biopsy-confirmed DN. None of these diabetic patients included have a family history of diabetes. We aimed to increase the homogeneity within the groups by including only male volunteers and patients. The urine extracellular vesicles of patients with DN, Type 2 diabetes, and healthy individuals were pooled for initial screening. Patients with DN were evaluated using quantitative phosphoproteomics to identify phosphoproteins with high or low abundance in uEVs. Three technical replicates were analyzed for each pooled sample.

### 4.2. Urine Collection and Pre-Processing

Midstream urine samples were collected in the morning and 10% NaN_3_ protease inhibitor (complete in EDTA; Roche Diagnostics) was added within 1 h. To minimize contamination, protein degradation, and incomplete collection, spot urine is preferred over 24-h urine samples. To remove cell debris, fresh urine samples were centrifuged at 4000× *g* for 15 min at 4 °C. After centrifugation, the supernatant was stored at −80 °C until further use.

### 4.3. uEVs Separation and Characterization

#### 4.3.1. Urinary Extracellular Vesicle Separation

As a part of the separation process, frozen samples were thawed and centrifuged for 20 min at 17,000× *g* and 4 °C. We used this procedure to pelletize protein precipitates caused by freeze-thaw cycles. After decanting the supernatant into polycarbonate ultracentrifugation containers (supplied by Beckman Coulter, Los Angeles, CA, USA), it was ultracentrifuged for 1 h at 4 °C at 150,000× *g*. Thereafter, we centrifuged the pellets at 200,000× *g* for 24 h at 4 °C on a 5–30% sucrose D_2_O gradient. We collected the sections with the lowest density from the upper part of the gradient as indicated by three visible bands and stored them at −80 °C. In this protocol, we departed from earlier studies on uEVs in Diabetic Kidney Disease (DKD), which used ultracentrifugation to enrich uEVs. A distinguishing feature of our method is the removal of urinary proteins via gradient density centrifugation, which mitigates their potential influence on phosphoproteomics.

#### 4.3.2. Transmission Electron Microscopy

The ultracentrifugation-derived pellet was reconstituted in 150 μL of 4% paraformaldehyde in PBS (pH 7.2) and subsequently incubated at 4 °C. Negative staining of the grid was performed by immersing it in 20 μL of 2% uranyl acetate solution in distilled water for 1 min. A 15-μL sample was spotted onto parafilm, and the grid was placed on 15-μL of distilled water for 5 min. To facilitate drying, the grid was set on filter paper for approximately 10 min. Finally, images were acquired using a 120 KV FEI Tecnai G2 Spirit transmission electron microscope.

#### 4.3.3. Nanoparticle Tracking Analysis

The NanoSight NS300 tool, along with NTA-3.4 software (Malvern Panalytical, Malvern, UK), was used to evaluate the size distribution and concentration of urinary EV samples. This equipment incorporates a blue laser module operating at a wavelength of 488 nm, a temperature-regulated flow-cell plate, and a single syringe. To ensure the optimal particle concentration for the final readings, which should be between 1.0 × 10^6^ and 1.0 × 10^9^ particles per milliliter, the samples were diluted with ultrapure water sourced from Thermo (located in Manassas, VA, USA). Five standard tests, each lasting 1 min, were conducted at a consistent temperature of 25 °C, with an automatic circulation system. The camera level was consistently set at 12 throughout all the sample assessments, and the detection threshold was fixed at 5.

#### 4.3.4. Western Blot Analysis

The uEVs were resuspended and diluted in a buffer that comprised 0.5% SDS and 1.1% Triton X-100. The process of size separation was undertaken through SDS PAGE gel electrophoresis. Afterward, the proteins were transferred onto polyvinylidene difluoride membranes and probed with specific antibodies, including anti-ALIX (Santa Cruz sc-53540), anti-TSG101 (Santa Cruz sc-7964), and anti-CD63 (Santa Cruz sc-5275). The testing was enhanced by verifying the samples using an anti-AQP2 (phospho S256) antibody (abcam 111346) and an anti-GSK3β (phosphor Y216) antibody (abcam 4797). Enhanced chemiluminescence facilitated the visualization of these tests.

### 4.4. Protein Digestion

The uEVs, which were preserved at −80 °C, were subjected to two washes using 20 mL of PBS. Subsequently, the resulting pellets were lysed with a solution that included 3 mL of 8 M urea, 30 μL of 100 mM ammonium bicarbonate, and 30 μL of a phosphatase inhibitor cocktail supplied by Sigma-Aldrich, St. Louis, MO, USA. Based on the protein concentration calculated via the BCA method, dithiothreitol was introduced until it attained a final concentration of 10 mM, and the solution was incubated at 60 °C for half an hour. The subsequent step entailed reduction through alkylation with iodoacetamide, which was adjusted to a final concentration of 20 mM and kept at room temperature in a dark setting for 30 min. Additional dithiothreitol was introduced to offset the surplus iodoacetamide until a final concentration of 20 mM was reached. This mixture was then diluted with 0.1 M ammonium bicarbonate to a final volume of 1.2 mL. Trypsin, in the amount of 150 μg at a 1:50 enzyme-to-sample ratio, was incorporated, and the solution was left to digest overnight at 37 °C. The resulting peptide solution was acidified with 1.5% TFA, desalted using a PS/DVB reversed-phase (RP) extraction column, and eluted with a 25% acetonitrile solution containing 0.1% TFA. The peptide portions were then lyophilized through vacuum centrifugation and stored at −80 °C.

### 4.5. Phosphopeptide Enrichment

A 50% suspension of Ni-NTA magnetic resin was utilized in this experiment. We started with a triple rinse of one milliliter of Ni-NTA beads using 425 μL of water. Following this, the beads underwent treatment with 400 μL of 100 mM EDTA (at pH 8.2) for half an hour, employing a technique of end-over-end rotation. Post-treatment, the EDTA solution was discarded and the beads were subjected to three additional rinses with 400 μL of water. The beads then underwent a 30-min treatment with 600 μL of a 10 mM FeCl3 water solution, using the same rotation method. To eliminate excess metal ions, the beads were again rinsed three times with 430 μL of water and subsequently resuspended in a mixture of methanol, 0.01% acetic acid, and acetonitrile in equal parts. The trypsin-digested peptide was mixed with the beads in an 80% MeCN/0.1% TFA solution, allowing for the phosphopeptides to be captured through 30 min of end-over-end rotation. The supernatant was then discarded, and the beads were washed three times with 400 μL of 80% acetonitrile/0.1% TFA. The phosphopeptides were subsequently eluted using 50 μL of a 1:20 ammonia/water solution for 30 min and dried down to 5 μL via vacuum centrifugation. Lastly, the samples were rehydrated in a 20 mM ammonium formate buffer.

### 4.6. Three-Dimensional RP-SAX-RP Systems

Utilizing an isocratic pump, a range of ion pairing agents were administered to a 150 μm I.D. capillary filled with 5 μm of C18 resin. The running buffer was consistently held at a pH of 10.0 across the initial analytical column in the first dimension. Due to the adjustments made to the trapping and vent valves, the flow was channeled toward the secondary dimension pre-column. At the same time, an isocratic pump supplied a 1.2 μL/min flow of pH10 buffer to support the transfer of peptides between the two-dimensional columns. A first-dimension reversed-phase (RP) column outlet had an anion exchange column connected to it. This column was composed of 150 μm I.D. capillaries packed with 5 cm of SAX-NP5 resin, which was used for peptide elution with acetonitrile fraction injections maintained at a pH of 10.0. The peptides were then eluted from the column using varying quantities of ammonium formate and potassium chloride, also injected at a pH of 10.0. A trapping and vent valve directed the flow to the pre-column in the third-dimension column to transfer peptides between the second- and third-dimension columns. For 35.5 min, the isocratic pump provided 1.2 μL/min of ammonium formate buffer at pH10.0 and a binary pump provided 10 μL/min of 0.2 M acetic acid (to dilute the organic content and acidify the column effluent). Peptides were eluted by injecting acetonitrile fractions at pH 10.0, with or without ion-pairing agents. Both valve positions were altered post-trapping to create a split in the column, allowing the gradient elution (0.5–40% B in 60 min, A = 0.2 M acetic acid, B = acetonitrile with 0.2 M acetic acid) of peptides to a column (25 m I.D. capillary containing 50 cm of 5 mm RB C18 resin (Sepax, Newark, DE, USA) with an integrated 1mm emitter tip), and into a mass spectrometer at a flow rate of 3–5 nL/min. The experimental design ensured the transfer of equivalent peptide complexities to the final column with each injection. Consequently, the organic eluents used to elute peptides from the first to the second dimension typically contained low salt concentrations (70 mM ammonium formate or 10 mM KCl, pH 10), so that a subset of peptides could be simultaneously eluted to the final dimension column. During gradient elution, a binary pump supplied 10 μL/min of 0.1% formic acid to dilute the organic content and acidify the effluent from the previous two dimensions. Furthermore, at a flow rate of 4–5 nL/min, the binary pump delivered acetonitrile and formic acid to the final column to elute the peptides. The fractionation experiment was completed, and the columns were regenerated with 90% acetonitrile, 1 M KCl in 10% acetic acid, 50% acetonitrile with 500 mM ammonium formate, and 900 mM ammonium formate in 10% acetic acid. In addition, three rapid organic gradients of 5 min were applied to the final dimension column.

### 4.7. LC-MS/MS Analysis

The analysis of the top 50 most abundant precursor ions from each MS scan was carried out using a Sciex 5600 mass spectrometer (Sciex, Framingham, MA, USA) in the data-dependent acquisition mode. Six parameters guided this procedure: collision-activated dissociation (CAD), electron multiplier detection, 35% collision energy, an isolation width of 2.0 Da, and a threshold of 200. The ESI voltage was configured to 2.4 kV, and the MS spectra were logged for a duration of 0.5 s. Dynamic exclusion was activated with a repeat count of 1 and an exclusion duration of 30 s. The MS/MS data analysis utilized a minimum threshold of 50 counts, a charge state range of 2+ to 4+, and a multiplier value of 4. In every cycle, up to 50 precursors were chosen, with a 20-s exclusion period after MS/MS selection. A broad CAD range (*m*/*z*: 140–1400Da) was established for each precursor. ESI sources from PicoView facilitated the precise positioning of the emitter tip at the mass spectrometry orifice for every analysis. The emitter tip was consistently rinsed with water while loading the sample or trapping peptides on the final pre-column.

### 4.8. Data Processing

#### 4.8.1. Data Processing

The default settings of the multiplier script wiff_2_mgf.mz were adopted. The TripleTOF 5600 data files were directly accessed and converted into mgf format for each experiment. The files were then searched with Mascot version 2.3 against a forward-reversed Swiss-Prot database (22,318 forward entries) along with a 643-entry cRAP database (the common repository of adventitious proteins (cRAP) database. Ion tolerances were set at 0.6 and 0.1 Da for the precursor and product ions, respectively. The search parameters, in addition to trypsin specificity, included carbamidomethylation (C, 57 Da), variable deamidation (NQ, 1 Da), oxidation (M, 16 Da), phosphorylation (STY, 80 Da), and fixed iTRAQ modification (N-term, K 145 Da). A spreadsheet encapsulating Mascot search results was crafted using the search results. Reverse database hits were removed using FDR_filter.mz and forward hits with a false discovery rate (FDR) higher than 1.0% were marked. A list of unique peptides (FDR 1%) from each analysis was generated using Multiplierz’s multifile detection feature, and phosphorylated peptides were tallied using a script. Consequently, unique phosphorylated peptides can be distinguished by merging their sequences and phosphorylation states. Sequences phosphorylated at diverse residues were counted once; however, sequences with different counts of phosphorylation sites were enumerated separately.

#### 4.8.2. Phosphorylation Site Localization

We employed the MD-score approach to assess the localization of phosphorylation sites, facilitated by a multiplierz script. The MD score of a phosphopeptide is derived from the Mascot score difference between its top-ranking and second-top-ranking peptide hits having the same sequence but varying phosphorylation sites. In instances where no such hits existed, an MD score was utilized. We adopted a cutoff value of 10 for these data to measure the phosphorylation sites at the protein level.

#### 4.8.3. Quantitative Data Analysis

The peak intensity values of the iTRAQ reporter ions were derived from the CAD scans (reporter_iTRAQ.mz) and adjusted for recognized isotopic contaminants (iTRAQ_software_corrections.mz (version V1.05)). Independent LC-MS/MS analyses of the Fe3-NTA resin supernatant were conducted, and the signals from every quantitation channel (114, 115, 116, and 117) were summed (across all peptides) to calculate normalization factors and correct the phosphopeptide data for minor variations in the source protein concentrations. The iTRAQ reporter signal intensities from the peptides, derived from several MS/MS scans, were accumulated prior to ratio calculation The “error model” incorporated into the Multiplierz script was employed to calculate the proportion of identified phosphorylated peptides between groups and the p-value of significant changes. Phosphorylated peptides displaying a change factor above 1.5 times and a p-value below 0.05 were identified.

### 4.9. Immunohistochemistry Stain

Kidney sections from mouse models and patients with DN, embedded in paraffin, were deparaffinized and rehydrated. An extensive outline of the DB/DB model has been documented recently. Periodic acid-Schiff staining was conducted in accordance with established protocols. Urinary and kidney samples from patients with DN were histochemically evaluated following biopsies displaying typical pathological alterations. The diagnosis of IgA nephropathy in diabetic patients necessitates kidney samples for histochemical examination. An incubation process with peroxide was employed to neutralize the endogenous peroxidase, which was succeeded by a blocking step for 1 h using 10% normal goat serum in PBS. The sections were treated with the primary rabbit polyclonal anti-AQP2 (phospho S256) antibody (at a 1:100 dilution) and primary rabbit polyclonal anti-GSK3β (phospho Y216) antibody (at a 1:50 dilution), and left overnight at 4 °C. The sections were then allowed to incubate at room temperature for 1 h, washed with PBS, and treated for 10 min with a secondary biotinylated mouse anti-polyvalent antibody. After undergoing four PBS washes, the sections were treated with streptavidin peroxidase for 10 min at room temperature, rinsed three times in PBS, and developed using a diaminobenzidine chromogen substrate for 3 min. Hematoxylin was used as a counterstain, and this was followed by rehydration and slide mounting.

## 5. Conclusions

This study aimed to identify reliable biomarkers of diabetic nephropathy (DN) using quantitative phosphoproteomics to analyze urinary extracellular vesicles (uEVs). In total, 233 phosphopeptides, including those detected by mass spectrometry, were detected within the uEVs. Forty-seven (47) phosphoproteins exhibiting significant alterations in patients with DN compared to those in patients with diabetes were identified. From these phosphoproteins, phosphorylated aquaporin-2 (p-AQP2[S256]) and phosphorylated glycogen synthase kinase-3β (p-GSK3β[Y216]) were selected for validation, as previously implicated in DN pathogenesis. This study confirmed the alterations of these phosphoproteins in uEVs through Phos-tag western blotting and immunohistochemical staining of kidney tissue, suggesting their potential as highly feasible diagnostic tools for kidney disease.

## Figures and Tables

**Figure 1 molecules-28-05605-f001:**
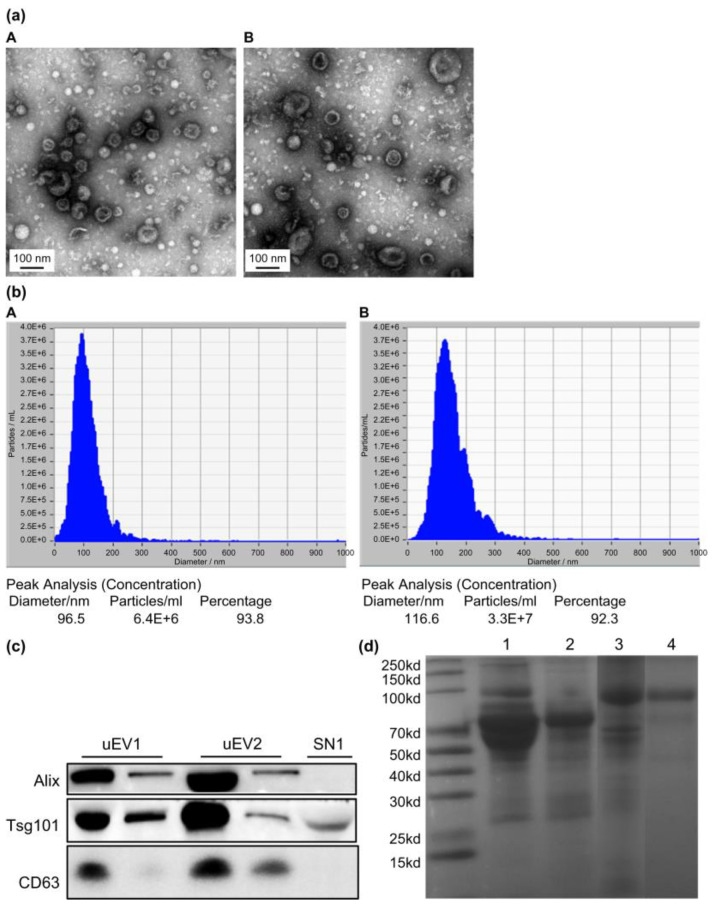
Characterization of uEVs isolated by the ultracentrifugation method and sucrose gradient density centrifugation. (**a**) Transmission electron microscopy of isolated uEVs depicts uEVs of typical shape and size ((**A**): uEVs isolated by ultracentrifugation; (**B**): uEVs isolated by sucrose gradient density centrifugation). (**b**) Size distribution of isolated uEVs by Nanoparticle Tracking ((**A**): uEVs isolated by ultracentrifugation; (**B**): uEVs isolated by sucrose gradient density centrifugation). (**c**) Quality assessment of isolated uEVs by western blot analysis for uEVs markers (uEV1: uEVs isolated by sucrose gradient density centrifugation; uEV2: uEVs isolated by ultracentrifugation; SN1: supernatant obtained by ultracentrifugation). (**d**) SDS-PAGE gel electrophoresis of uEVs from NC subjects and DN patients showing the coprecipitation albumin with uEVs by ultracentrifugation can be removed by sucrose gradient density centrifugation (Lane 1, uEVs from DN patients with large amounts of urine proteins isolated by ultracentrifugation; Lane 2, uEVs from DN patients with large amount of urine proteins isolated by sucrose gradient density centrifugation; Lane 3, uEVs from NC subjects without proteinuria isolated by ultracentrifugation; Lane 4, uEVs from NC subjects without proteinuria isolated by sucrose gradient density centrifugation).

**Figure 2 molecules-28-05605-f002:**
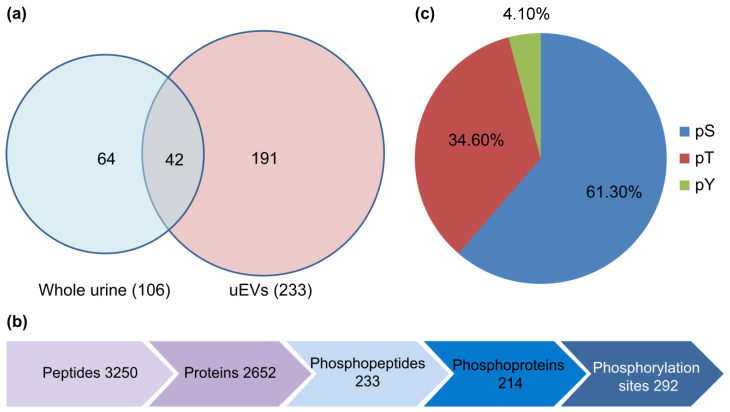
(**a**) The Venn diagram showing the number of unique phosphopeptides identified in urine and uEVs. (**b**) The diagram shows the number of proteins, phosphopeptides, phosphoproteins, and phosphorylation sites identified in uEVs. (**c**) The distribution of serine/threonine/tyrosine (pS/pT/pY) phosphopeptides in uEVs.

**Figure 3 molecules-28-05605-f003:**
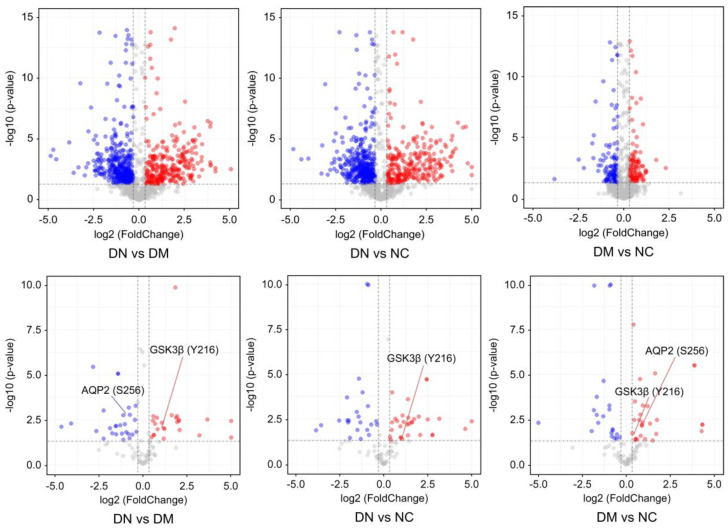
The volcano plots represent the quantitative analysis of the proteomes and phosph-proteomes of uEVs from patients with DN, DM, and NC subjects (DN = Diabetic nephropathy; DM = diabetes; NC = normal control). The colored circles indicates the phosphopeptides detected significantly changed among patients with DN, DM, and NC subjects. The red and blue circles indicates the phosphopeptides detected significantly upregulated and downregulated, respectively.

**Figure 4 molecules-28-05605-f004:**
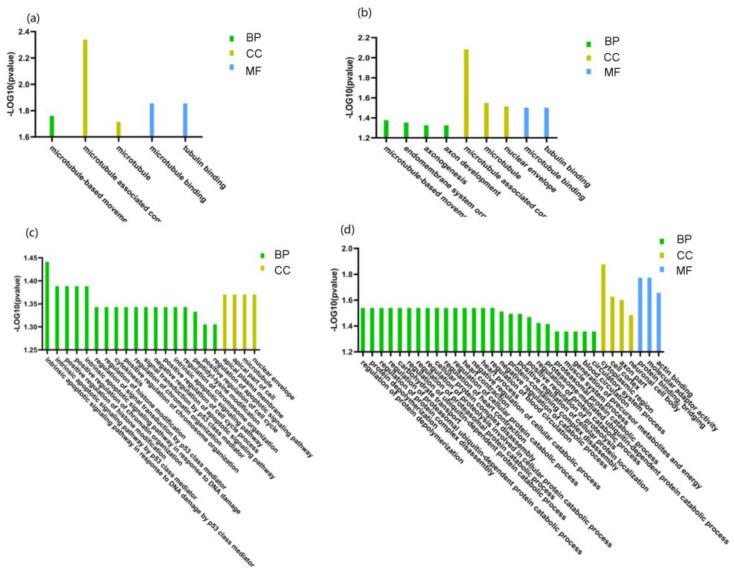
Gene Ontology (GO) enrichment analysis of significantly changed phosphoprotein in uEVs among patients with DN, DM, and NC subjects. (**a**) GO enrichment analysis of significantly upregulated phosphoproteins of uEVs from DN patients compared to DM patients. (**b**) GO enrichment analysis of significantly upregulated phosphoproteins of uEVs from DM patients compared to NC subjects. (**c**) GO enrichment analysis of significantly upregulated phosphoproteins of uEVs from DM patients compared to NC subjects. (**d**) GO enrichment analysis of significantly downregulated phosphoproteins of uEVs from DM patients compared to NC subjects. (BP: biological process; CC: cell component; MF: molecular function).

**Figure 5 molecules-28-05605-f005:**
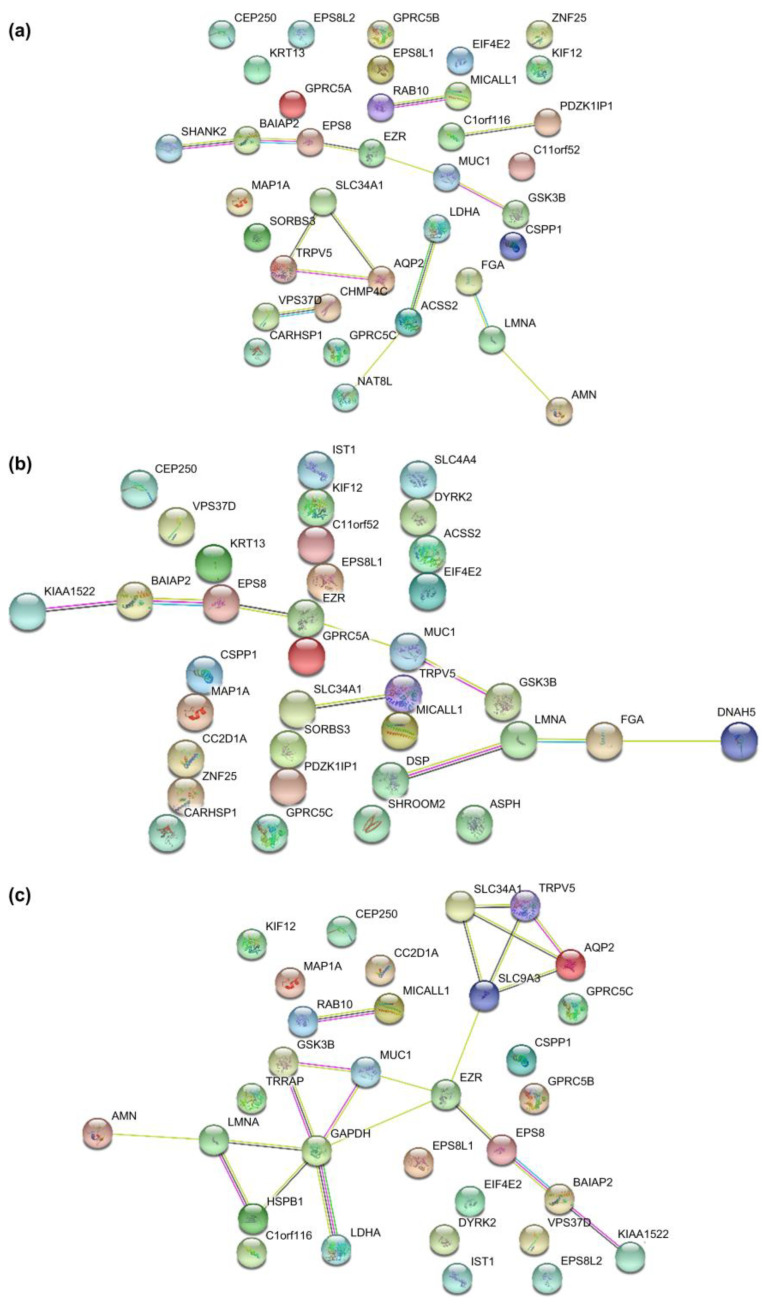
The interaction network of aberrant phosphoproteins in uEVs among patients with DN, DM, and NC subjects was constructed by STRING at confidence score >0.7. (**a**) The STRING network analysis of aberrant phosphoproteins in EVs from DN patients compared to DM patients. (**b**) The STRING network analysis of aberrant phosphoproteins in EVs from DN patients compared to NC subjects. (**c**) The STRING network analysis of aberrant phosphoproteins in uEVs from DN patients compared to NC subjects.

**Figure 6 molecules-28-05605-f006:**
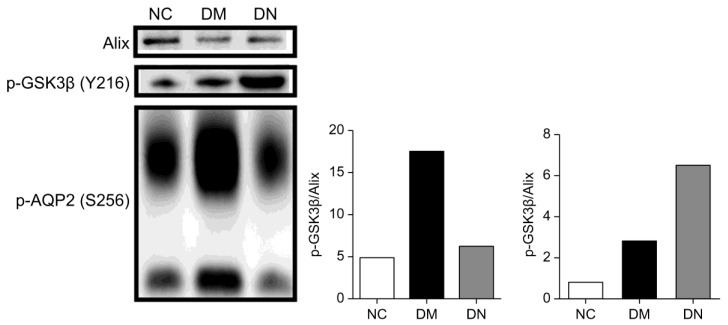
Phos-tag western blotting comparing p-AQP2(S256) and p-GSK3β(Y216) in uEVs from patients with DN, DM, and NC subjects for confirmation. The Histograms represent the normalized phosphoproteins p-AQP2(S256)/Alix and p-GSK3β(Y216)/Alix.

**Figure 7 molecules-28-05605-f007:**
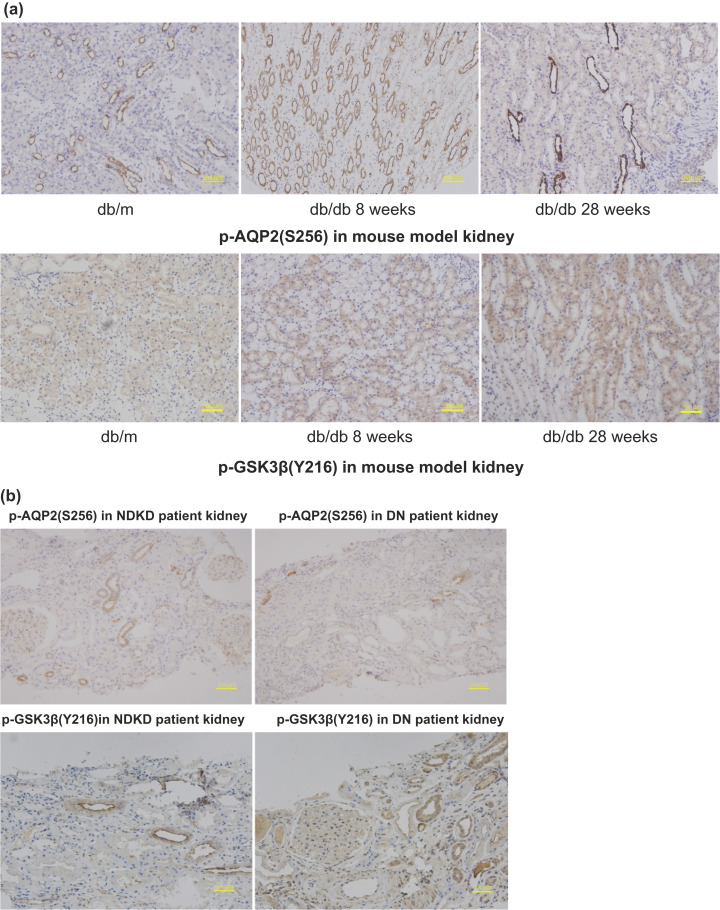
(**a**) PAS staining in kidney tissue of db/m, db/db 8 w, and db/db 28 w to show the typical pathological changes of DN in db/db 28 w compared to db/db 8 w with no pathological changes of DN. (**b**) Immunohistochemistry staining of p-AQP2(S256) and p-GSK3β(Y216) in kidney tissue of db/db 8 w, db/db 28 w and db/m, respectively.

**Table 1 molecules-28-05605-t001:** Clinical characteristics of the three subject groups.

Groups	Age (Years)	HbA1c (%)	ACR (μg/mg)	eGFR (mL/min/ 1.73 m^2^)	BUN (mmol/L)	Scr (μmol/L)
NC	50.20 ± 4.53	NA	NA	109 ± 9.08	5.62 ± 0.31	66.8 ± 5.49
DM	55.80 ± 9.20	6.70 ± 0.24 *	83.68 ± 6.13 *	113.40 ± 5.43 **	6.56 ± 0.40 *	81.6 ± 9.91 **
DN	59. 40 ± 7.74	9.46 ± 1.75	2418.64 ± 1478.57	47.80 ± 14.57	8.62 ± 1.49	151 ± 28.35

(NC = normal control; DN = diabetic nephropathy; DM = diabetes; HbA1c = Hemoglobin A1C; ACR = albumin-to-creatinine ratio; eGFR = estimated glomerular filtration rate. * *p* < 0.05 vs. DN group; ** *p* < 0.001 vs. DN group).

**Table 2 molecules-28-05605-t002:** uEVs phosphopeptides significantly changed among patients with DN, DM, and NC subjects. (**a**) Significantly upregulated phosphorylated peptides of uEVs from DN patients compared to DM patients. (**b**) Significantly downregulated phosphorylated peptides of uEVs from DN patients compared to DM patients. (**c**) Significantly upregulated phosphorylated peptides of uEVs from DN patients compared to NC subjects. (**d**) Significantly downregulated phosphorylated peptides of uEVs from DN patients compared to NC subjects. (**e**) Significantly upregulated phosphorylated peptides of uEVs from DM patients compared to NC subjects. (**f**) Significantly downregulated phosphorylated peptides of uEVs from DM patients compared to NC subjects.

(a)			
Gene	Peptide	Phosphor_Site	*p*-value
ACSA	SWSPPPEVSR	S3:Phospho	0.00240942
AMNLS	RLSLVPK	S3:Phospho	0.01164248
BAIP2	AFPAQTASGFK	T6:Phospho	0.003610772
BAIP2	LSDSYSNTLPVR	T8:Phospho	0.00927998
CHSP1	GNVVPSPLPTR	S6:Phospho	0.011014319
ES8L1	SLNSTPPPPPAPAPAPPPALAR	T5:Phospho	0.031333883
FIBA	PGSTGTWNPGSSER	S11:Phospho	0.003771194
GSK3B	GEPNVSYICSR	Y7:Phospho	0.028989048
GPC5B	TAGFPNGSLGK	S8:Phospho	0.022155551
GPC5C	ATANSQVMGSANSTLR	S13:Phospho	0.010074399
K1C13	MIGFPSSAGSVSPR	S12:Phospho	0.003025495
KIF12	SPGQVLPPH	S1:Phospho	1.37 × 10^−10^
MAP1A	TLPQEPGK	T1:Phospho	0.00176674
MILK1	SPVPSPGSSSPQLQVK	S10:Phospho	0.022496032
MY15B	AVPSPPPPPIVK	S4:Phospho	0.001997071
NAT8L	MSVDSRFR	S5:Phospho	0.029409743
NPT2A	LGSPAVSPLPVR	S7:Phospho	0.002418861
PDZ1I	YSSMAASFR	S7:Phospho	0.004247949
SHAN2	RAPSPVVSPTEMNK	S4:Phospho	0.035395595
VINEX	AIETRLPSPK	T4:Phospho	0.003764395
VP37D	SAQPAPTSAADPPK	S1:Phospho	0.023114006
YG015	IELSYNK	Y5:Phospho	0.002020309
ZNF25	SHFIIHQR	S1:Phospho	0.003312002
(b)			
AQP2	RQSVELHSPQSLPR	S3:Phospho	0.001562435
AQP2	RRQSVELHSPQSLPR	S4:Phospho	0.001865546
CAN7L	INSAHGSDKSK	S3:Phospho	0.005547201
CF170	KWMHAHYSR	S8:Phospho	8.55 × 10^−6^
CHM4C	LPNVPSSSLPAQPNR	S8:Phospho	0.03478666
CI169	NPYAHISIPR	S7:Phospho	0.014805139
CK052	TLKPQPQQLQQNLPK	T1:Phospho	0.015894205
CP250	QSESLSELITLR	S2:Phospho	0.007130084
CSPP1	QPSPIVPALQNK	S3:Phospho	0.000997041
EPS8	APAPAPPGTVTQVDVR	T9:Phospho	0.001709885
EPS8	KGPGEGVLTLR	T9:Phospho	0.003312002
ES8L2	HSPTSEPTPPGDALPPVSSPHTHR	S19:Phospho	0.041726109
EZRI	QLLTLSSELSQAR	T4:Phospho	3.66 × 10^−6^
GPC5C	VPSEGAYDIILPR	S3:Phospho	0.00052554
ICK	STPGLIPRPPAAQPVHGR	T2:Phospho	0.021340709
IF4E2	TASDQATTARIR	S3:Phospho	0.019186167
LDHA	TPKIVSGK	T1:Phospho	0.020570378
LMNA	SGAQASSTPLSPTRITR	S11:Phospho	0.00927998
MUC1	DTYHPMSEYPTYHTHGR	S7:Phospho	0.035395595
RAB10	FHTITTSYYR	T3:Phospho	0.006517696
RAI3	AHAWPSPYKDYEVK	S6:Phospho	0.01798913
SARG	ANSALTPPKPESGLTLQESNTPGLR	T6:Phospho	0.019581944
SHS6L	TPNLDWR	T1:Phospho	0.000684591
TRPV5	ASLALPTSSLSR	S11:Phospho	0.00176674
(c)			
Gene	Peptide	Phosphor_Site	*p*-value
ACSA	SWSPPPEVSR	S3:Phospho	0.00240942
AMNLS	RLSLVPK	S3:Phospho	0.01164248
BAIP2	AFPAQTASGFK	T6:Phospho	0.003610772
BAIP2	LSDSYSNTLPVR	T8:Phospho	0.00927998
CHSP1	GNVVPSPLPTR	S6:Phospho	0.011014319
ES8L1	SLNSTPPPPPAPAPAPPPALAR	T5:Phospho	0.031333883
FIBA	PGSTGTWNPGSSER	S11:Phospho	0.003771194
GSK3B	GEPNVSYICSR	Y7:Phospho	0.028989048
GPC5B	TAGFPNGSLGK	S8:Phospho	0.022155551
GPC5C	ATANSQVMGSANSTLR	S13:Phospho	0.010074399
K1C13	MIGFPSSAGSVSPR	S12:Phospho	0.003025495
KIF12	SPGQVLPPH	S1:Phospho	1.37 × 10^−10^
MAP1A	TLPQEPGK	T1:Phospho	0.00176674
MILK1	SPVPSPGSSSPQLQVK	S10:Phospho	0.022496032
MY15B	AVPSPPPPPIVK	S4:Phospho	0.001997071
NAT8L	MSVDSRFR	S5:Phospho	0.029409743
NPT2A	LGSPAVSPLPVR	S7:Phospho	0.002418861
PDZ1I	YSSMAASFR	S7:Phospho	0.004247949
SHAN2	RAPSPVVSPTEMNK	S4:Phospho	0.035395595
VINEX	AIETRLPSPK	T4:Phospho	0.003764395
VP37D	SAQPAPTSAADPPK	S1:Phospho	0.023114006
YG015	IELSYNK	Y5:Phospho	0.002020309
ZNF25	SHFIIHQR	S1:Phospho	0.003312002
(d)			
AQP2	RQSVELHSPQSLPR	S3:Phospho	0.001562435
AQP2	RRQSVELHSPQSLPR	S4:Phospho	0.001865546
CAN7L	INSAHGSDKSK	S3:Phospho	0.005547201
CF170	KWMHAHYSR	S8:Phospho	8.55 × 10^−6^
CHM4C	LPNVPSSSLPAQPNR	S8:Phospho	0.03478666
CI169	NPYAHISIPR	S7:Phospho	0.014805139
CK052	TLKPQPQQLQQNLPK	T1:Phospho	0.015894205
CP250	QSESLSELITLR	S2:Phospho	0.007130084
CSPP1	QPSPIVPALQNK	S3:Phospho	0.000997041
EPS8	APAPAPPGTVTQVDVR	T9:Phospho	0.001709885
EPS8	KGPGEGVLTLR	T9:Phospho	0.003312002
ES8L2	HSPTSEPTPPGDALPPVSSPHTHR	S19:Phospho	0.041726109
EZRI	QLLTLSSELSQAR	T4:Phospho	3.66 × 10^−6^
GPC5C	VPSEGAYDIILPR	S3:Phospho	0.00052554
ICK	STPGLIPRPPAAQPVHGR	T2:Phospho	0.021340709
IF4E2	TASDQATTARIR	S3:Phospho	0.019186167
LDHA	TPKIVSGK	T1:Phospho	0.020570378
LMNA	SGAQASSTPLSPTRITR	S11:Phospho	0.00927998
MUC1	DTYHPMSEYPTYHTHGR	S7:Phospho	0.035395595
RAB10	FHTITTSYYR	T3:Phospho	0.006517696
RAI3	AHAWPSPYKDYEVK	S6:Phospho	0.01798913
SARG	ANSALTPPKPESGLTLQESNTPGLR	T6:Phospho	0.019581944
SHS6L	TPNLDWR	T1:Phospho	0.000684591
TRPV5	ASLALPTSSLSR	S11:Phospho	0.00176674
(e)			
Gene	Peptide	Phosphor_Site	*p*-value
AQP2	RQSVELHSPQSLPR	S3:Phospho	0.00351656
AQP2	RRQSVELHSPQSLPR	S4:Phospho	0.003246139
CAN7L	INSAHGSDKSK	S3:Phospho	0.007011764
CF170	KWMHAHYSR	S8:Phospho	3.21 × 10^−6^
CI169	NPYAHISIPR	S7:Phospho	0.003453563
CP250	QSESLSELITLR	S2:Phospho	0.00612397
CSPP1	QPSPIVPALQNK	S3:Phospho	0.004669614
EPS8	APAPAPPGTVTQVDVR	T9:Phospho	0.000581478
EPS8	KGPGEGVLTLR	T9:Phospho	0.00052707
ES8L1	AAGEGLLTLR	T8:Phospho	1.84 × 10^−5^
ES8L1	SLNSTPPPPPAPAPAPPPALAR	S4:Phospho	0.014008085
ES8L2	HSPTSEPTPPGDALPPVSSPHTHR	S19:Phospho	0.040008069
EZRI	QLLTLSSELSQAR	T4:Phospho	8.73 × 10^−6^
GPC5C	GVGYETILK	T6:Phospho	1.68 × 10^−8^
GPC5C	VPSEGAYDIILPR	S3:Phospho	0.000326668
GSK3B	GEPNVSYICSR	Y7:Phospho	0.008989048
HSPB1	GPSWDPFR	S3:Phospho	0.047463946
ICK	STPGLIPRPPAAQPVHGR	T2:Phospho	0.006484348
LDHA	TPKIVSGK	T1:Phospho	0.020666271
MUC1	DTYHPMSEYPTYHTHGR	T11:Phospho	0.025694245
RAB10	FHTITTSYYR	T3:Phospho	0.005397277
SARG	ANSALTPPKPESGLTLQESNTPGL	RT6:Phospho	0.041546421
SHS6L	TPNLDWR	T1:Phospho	0.003375752
TRRAP	LEPAFLSGLR	S7:Phospho	0.043482561
YG015	IELSYNK	Y5:Phospho	0.001729708
(f)			
AMNLS	RLSLVPK	S3:Phospho	0.014008085
BAIP2	AFPAQTASGFK	T6:Phospho	0.00053512
C2D1A	GPASTPTYSPAPTQPAPR	S9:Phospho	1.85 × 10^−11^
DYRK2	KPSAAAPAAYPTGR	S3:Phospho	0.001809625
G3P	GALQNIIPASTGAAK	S10:Phospho	0.022036915
GPC5B	TAGFPNGSLGK	S8:Phospho	0.011034256
IF4E2	TASDQATTARIR	S3:Phospho	0.012911956
IST1	NISSAQIVGPGPKPEASAK	S3:Phospho	2.28 × 10^−5^
K1522	ASPVPAPSSGLHAAVR	S2:Phospho	0.017621205
KIF12	DLLSLGSPR	S7:Phospho	0.000979657
KIF12	SPGQVLPPH	S1:Phospho	1.17 × 10^−10^
LMNA	SGAQASSTPLSPTRITR	S11:Phospho	0.033756475
MAP1A	TLPQEPGK	T1:Phospho	0.037174836
MILK1	SPVPSPGSSSPQLQVK	S10:Phospho	0.002110621
MY15B	AVPSPPPPPIVK	S4:Phospho	0.02828527
NPT2A	LGSPAVSPLPVR	S7:Phospho	0.00017686
SL9A3	RGSLAFIR	S3:Phospho	0.031611131
TRPV5	ASLALPTSSLSR	S11:Phospho	0.000897584
VP37D	SAQPAPTSAADPPK	S1:Phospho	0.004954683

## Data Availability

The phosproteomic data of urinary extracellular can be accessed by contacting the corresponding author.

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
