# Peer review of "Phosphoproteome Profiling of uEVs Reveals p-AQP2 and p-GSK3β as Potential Markers for Diabetic Nephropathy"

_molecules, 2023, doi:10.3390/molecules28145605_

Round 1

Reviewer 1 Report

Overall, this appears to be a well-conducted study with significant potential implications for the diagnosis and monitoring of DN. The authors have demonstrated an innovative and powerful approach to identifying potential biomarkers and provided a foundation for future research in this area. Further work is needed to validate these findings in larger and more diverse patient populations, and to understand the potential clinical applications of these findings. The manuscript is generally well-written and clear, but additional detail and context in certain areas would enhance the paper. We suggest that this manuscript could be accepted in its current form or with minor revisions.

1.     The discussion is largely appropriate and ties the results back to the initial objectives of the study. The authors have done a good job acknowledging the limitations of their study and suggest steps for future research. The discussion would benefit from a broader examination of how these findings might be translated into clinical practice. Also, it might be interesting for the authors to discuss how their findings compare to other recent studies in the field.

2.     This appears to be a well-conducted study with significant potential implications for the diagnosis and monitoring of DN. The authors have demonstrated an innovative and powerful approach to identifying potential biomarkers and provided a foundation for future research in this area. Further work is needed to validate these findings in larger and more diverse patient populations, and to understand the potential clinical applications of these findings.

3.     The results are presented clearly and succinctly. The identification of such a large number of phosphoproteins in uEVs is an impressive achievement and a testament to the sensitivity and potential of their methodologies. More extensive exploration and discussion around the potential significance of the key phosphoproteins (p-AQP2(S256) and p-GSK3β(Y216)) identified in the study would be beneficial. While the authors mention these proteins’ roles, the direct correlation between these biomarkers and the pathogenesis and progression of DN could be elaborated. 

Reviewer 2 Report

The paper is well written, and their work is well presented. It should be published after the minor revisions shown below.

Minor points

• Data from clinical analyzes of urea and creatinine are available for the groups, add to the table

• In the case of the DM group, there is a clinical history or family history of the disease.

• Add the stature and weight of the groups.

• Do not observe Figure 1. Add

• What is the sensitivity of the method.

• In addition to urine, the detection of proteins was tested in another fluid.

check punctuation marks and long sentences.
